# Revaluing Gender and Religion in the Anthropological Debate of the Anthropocene: A Critique on the Threefold Culture–Nature–Supernature Divide

Catrien Notermans * and Anke Tonnaer

Department of Anthropology and Development Studies, Radboud University, P.O. Box 9104, 6500 HE Nijmegen, The Netherlands; anke.tonnaer@ru.nl
* Correspondence: catrien.notermans@ru.nl

**Abstract:** This study argues that current anthropological research on human–nature relatedness lacks an explicit focus on gender and religion. It brings to the forefront that most current studies in Anthropocene anthropology that move away from anthropocentrism and towards studying more-than-human relatedness imply a disregard of gender that concerns both the 'human' and the 'non-human' in their mutual relationships. Presuming that the concept of sociality does not distinguish between human and nonhuman, the authors believe, however, that expressions of gender in more-than-human social relatedness cannot be denied. Simultaneously, they state that Anthropocene scholarship, by conceiving a secular future for humans restoring their relatedness with nature, is inclined to leave the 'supernature' out and to ignore experiences and embodied practices of enchantment in the modern world. By reviewing the feminist anthropological literature on the nature–culture divide and exploring the potential of enchantment as a way out of the secular condition of anthropology, the authors aim to restore a focus on gender and religion in anthropological Anthropocene scholarship while also transcending the threefold nature–culture–supernature divide. This review offers the theoretical prelude and introduction to the contributions of the Special Issue "Gender, Nature and Religious Re-enchantment in the Anthropocene".

**Keywords:** gender; nature; religion; enchantment; culture–nature–supernature divide; Anthropocene; anthropology





## 1. Introduction

In the capitalist nexus of economy–ecology, the dominant narrative on human–nature relatedness unceasingly supports and fosters a persistent wish for material wealth and the possibilities of seemingly limitless consumption that globalization and economic progress have to offer. This narrative derives from colonial, Christian and industrial ontologies that continue to legitimize violence against indigenous people and their natural environments (e.g., Gosh 2021; Patel and Moore 2016; Plumwood 2002a). As a result, we live in a natural world characterized by 'absence', in which nature is 'reduced to inert matter vulnerable to human manipulation and exploitation' (Haberman 2020, p. 239); a world, in short, that is largely disenchanted. However, awareness of the necessity to fundamentally change humans' impudent and exploitative attitude to nature is growing. It is said that we have entered the age of the Anthropocene, the geological epoch during which human activity has been the dominant influence on the climate and environment (e.g., Crutzen 2006).[1] Climate change and the ensuing ecological crises push us away from human-centered capitalist practices, impelling us to 'decolonize our relationships with nature' (Plumwood 2002a), and 'work out new ways to live with the earth' (Rose 2013, p. 93).

Academic scholarship on the Anthropocene responds to the urgency of planetary and environmental restoration by criticizing the Western rationalist, capitalist and Christian nature–culture divide that goes together with a self-evident anthropocentrism in scientific

traditions. By developing a posthuman perspective that integrates the more-than-human world in contemporary research projects, scholars increasingly focus on people's embodied and affective bonding with the natural world (e.g., Descola 2014; Hastrup 2014; Ingold 2002; Strang 2015), on multispecies entanglements and cohabitations (e.g., Tsing 2015; Van Dooren 2016) and on their imaginations thereof (Haraway 2016). Yet, as anthropologists and co-editors of this Special Issue, we believe that within this ontological turn, a distinct focus on gender and religion is hitherto missing.

We found that moving away from anthropocentrism and studying more-than-human relatedness in anthropology apparently implies a disregard of gender, a disregard that concerns both the 'human' and the 'non-human' in their mutual relationships. Assuming that 'nature is always implicit in the making of social life' (Hastrup 2014) and that 'the concept of sociality does not distinguish between human and non-human' (Tsing 2014, p. 27), we believe that expressions of gender in more-than-human social relatedness cannot be denied (see also: Notermans and Pfister 2016; Govindrajan 2018; Helmreich 2017; Hovorka 2012; Saugeres 2002). Simultaneously, we found that Anthropocene scholarship tends to focus on secular contexts, as well as to implicitly rely on secular worldviews, thus drawing and reproducing ontological borders, not only between 'culture' and 'nature', but also between 'nature' and 'supernature'. In conceiving a secular future for humans, restoring their relatedness with nature, this scholarship is inclined to leave the 'supernature' out, and to ignore experiences and practices of enchantment in the modern world. Of course we do acknowledge that interdisciplinary work exists with an ethnographic focus on gender and nature in religious contexts (e.g., Fedele and Knibbe 2013, 2020; Starkey and Tomalin 2022), yet these do not directly engage with the socio-ecological impact and the changing gendered relations the Anthropocene generates.

In this Special Issue, we therefore bring together a multidisciplinary collection of ethnographic studies that join the contemporary societal and scientific debates on the Anthropocene and the related process of climate change while deliberately bringing into focus the embodied practice of gender and religion. In doing so, we aim to transcend the threefold nature–culture–supernature divide, while also adding a gender-sensitive approach to the study of more-than-human entanglements. In the remaining part of this introduction we will elaborate on the theoretical background of studying "Gender, Nature, and Religious Re-Enchantment in the Anthropocene", and introduce the set of contributions we selected for this Special Issue.

## 2. Gender and Nature

To introduce our concern with the relationship between gender and nature, we need to go back to the origin of 'women studies' in anthropology. Central to the debate in early feminist anthropology was the understanding of women's subordinate position through the nature–culture divide. In a daring and committed paper, published in the volume *Women, Culture, Society* edited by Rosaldo and Lamphere (1974), the young feminist anthropologist Sherry Ortner sought to understand the universality of women's subordination by asking 'Is female to male as nature is to culture'? She wrote:

> My thesis is that woman is being identified with—or, if you will, seems to be a symbol of—something that every culture devalues, something that every culture defines as being of a lower order of existence than itself. Now it seems that there is only one thing that would fit that description, and that is "nature" in the most generalized sense. (Ortner 1974, pp. 71–72)

Ortner's provocative statement linked women and nature in their subjugation to men and culture. This triggered a lively debate, with the most intense criticism coming from feminist anthropologists. With the edited volume *Nature, Culture, and Gender* (MacCormack and Strathern 1980) Carol MacCormack and Marilyn Strathern criticized Ortner's proposition that worldwide, women would have a lower status in society because of their status of 'nature'. The universal character of the statement was criticized, yet it was mainly the woman–nature association that caused the most outrage. The very idea of a natural femi-

nine connection with nature openly situated women in the realm of animals rather than that of humans, which was felt as deeply humiliating. In fact, this feminist outrage confirmed Ortner's critical observation of women being subjugated as a result of their 'naturalness'.

The societal context at that time was colored by second-wave feminism, a liberation movement particularly concerned with freeing women from dominant patriarchal ideologies that reduced women to nature. Western liberal feminists considered women's status of 'naturalness'—of being biologically determined to reproduce and become nursing mothers—the main cause of women's lack of equal rights in society at that time. They thus fiercely contested the woman–nature connection and defended women's rights to transcend their allegedly passive status of nature, to become cultural agents equal to men, and to empower themselves by governing their bodies and reproductive capacities. Male-controlled societal expectations of women delivering unpaid housekeeping and unremitting care to husband and children were to be replaced by women's autonomy and right of self-determination.[2]

What matters here for our discussion is that feminist anthropology did *not* contest the nature–culture divide that positioned animals, the earth, and other nonhuman bodies at the lower nature side of the hierarchy and humans at the higher culture side. Supporting the ideals of liberal feminism, Western feminist scholars equally considered nature to be distinct from and inferior to culture, and thus rejected the low positioning of women in the (passive, emotional, and irrational) realm of nature. Or, in the words of MacCormack (1980, p. 1): 'The process of nature becoming culture is the process by which we feel we became human'. Women had to be included as *humans* into the same category as men and consequently gain more freedom, power and status in society. Completely in line with the nature–culture divide, yet in a fierce response to the biological reduction of women to a status of 'subordinate nature', feminist scholars then developed a corresponding sex–gender binary (e.g., Rosaldo and Lamphere 1974; Shapiro 1981). The separation of gender, as a social role, from sex, as the biological body, shifted the focus on *humans*' capacity (thus of *both* men *and* women) to pass from nature to culture (see also MacCormack 1980, p. 17).

The Western feminist claim for full humanity for women was uncritical to the human strategy of dominating and gaining control over nature. We believe this (partly) explains why early feminist anthropology, despite its focus on the nature–culture divide, does not resonate in the anthropological human–nature debate today as the latter deliberately aims to deconstruct and transcend the divide. Another reason why feminist scholarship stays away from the anthropological discussion on human–nature entanglements is the enduring liberal feminist reluctance to associate women with nature in feminist anthropology today (e.g., O'Reilly 2019). Critically reflecting on this persistent liberal feminist stand, ecofeminist Val Plumwood (1993, pp. 19–20) mentions that the result of it is that women's 'closeness to nature' is still hardly a compliment, because 'nature' as the excluded and devalued contrast of 'culture' would represent everything that 'reason' excludes: the body, emotions, animality, physicality, and the nonhuman world.

Plumwood's (1993) standpoint matters here as ecofeminism developed from ecology movements into another branch of feminism.[3] The term 'Ecofeminism' was coined by the French writer Francoise d'Eaubonne (1974), in the same year that Ortner (1974) published her classic feminist paper. In the late 1970s and the early 1980s, ecofeminists equally focused their discussion on the women–nature connection. Yet, they took a radically different position by claiming that 'arguments for women's freedom cannot convincingly be based on a similar putdown of the non-human world' (Plumwood 1993, p. 24). In solidarity with rather than in opposition to nature, ecofeminism still rejects the dualist approach and the assumed inferiority of *both* women *and* nature. The valuation of women's bodies and their embodied relationship with nature, as well as the parallel ecofeminists draw between the subjugation of women and nature in colonial contexts are crucially different from the standpoints in liberal feminist scholarship in anthropology (e.g., Gaard 1993; King 1990; Merchant 1995; Mies and Shiva 1993; Roa 2012; Shiva 1988). Even so, (eco)feminists' engagement with gender and nature is equally hardly reflected in contemporary anthropological attempts to think beyond the human–nature divide.[4]

Other (interdisciplinary) academic fields, however, are less reluctant to integrate a focus on gender in studying socialities beyond the human. Gender Studies turn to posthumanism and ecofeminism, explicitly *valuing* rather than contesting the affinity between women and nature, and stressing the *significance* of care for more-than-human worlds (e.g., Braidotti 2022; Haraway 2016; Lettow and Nessel 2022; Puig de la Bellacasa 2017). Moreover, Development Studies amply report that climate change and its related ecological threats particularly affect women as the main actors in nurturing and taking care of the environment, while also drawing attention to women's lack of power in global policy (e.g., Agarwal 1992; Arora-Jonsson 2011; Dankelman 2010; Denton 2002; Detraz 2023; MacGregor 2017; Vakoch and Mickey 2018). Out of the 14 contributions in the interdisciplinary volume *Feminist Ecologies* (2018), Deborah Bird Rose's (2018) work is the only anthropological one. Additionally, in the recently published volume *Ecologies of Gender* (Lettow and Nessel 2022), which explicitly starts from the assumption that 'gender relations are always at the same time nature relations, and nature relations must conversely be conceived as mediated by gender relations' (ibid.: 3), none of the 13 contributions are written from a feminist anthropological perspective. In discussing gender and its intersecting inequalities, anthropology thus strikingly stays behind.

In line with critical voices in Ecofeminist and Gender Studies, we do not naively engage in studying the *Man*thropocene in hu*man*–nature relationships (e.g., Di Chiro 2017). We rather aim to re-introduce gender as a central concept in studying our entanglements with nature, and by doing so, to revalue and revive the classical feminist scholarship in our disciplinary debate. We pick up the thread yet deliberately distance ourselves from liberal feminist binary thinking that devalues both nature and women's relatedness to it. We rather aim to give Ortner's classical study on the woman–nature connection a positive spin by combining the effort to transcend the nature–culture *and* the sex–gender divide with an ecofeminist revaluation of 'the area of culture and life that are associated with the feminine and with nature' (Plumwood 1993, p. 64). 'Both the sex-gender distinction and its deconstruction through the concept of gender,' Lettow and Nessel (2022, p. 3) state, 'have epistemologically downplayed the "natural" or "organic" aspects of human bodies as well as their interrelations with nonhuman creatures and materials.' This, we believe, needs to be rectified.

In this Special Issue, the ethnographic contributions thus intentionally focus on the lived experience of gendered human–nature relationships. We consider genders as plural, relational and inclusive of both male and female. We also assume that both humans and nonhumans are persons and thus gendered, yet express their genders differently in shifting relational contexts (e.g., Bird-David 2017; de Waal 2022). Gender thus *neither* refers to one particular biological sex (male or female or other) *nor* to any fixed gender identity, but rather to fluid and nonbinary interpretations of femininity and masculinity that may vary across natural bodies, both human and nonhuman. The contributions in this Special Issue seek to know how women and men perform their gender in relationship to a diversified more-than-human world and how both humans and nonhumans are gendered in their mutual relationship. We wonder if they find power in this relatedness, and how that relates to larger societal structures of dominance and inequality. Together the contributions will, in the wording of Sherilyn MacGregor (2017, p. 6), represent 'an inclusive framework for understanding the diversity of ways gender is performed in human and other species, and how these ways shape and are shaped by relations to the environment'.

## 3. Religion and Nature

In this Special Issue, we combine our effort to bring gender back into the nature–culture debate, with an attempt to restore a focus on religion in anthropological Anthropocene scholarship. We aim to resist the 'secular constitution of anthropology' (Furani and Robbins 2021) and to acknowledge that thinking beyond the human is also thinking beyond the secular. Together with the nature–culture divide, we thus seek to transcend the nature–supernature divide and explore how anthropological studies on more-than-human entanglements may

open up to non-secular ontologies that consider immaterial and invisible more-than-human powers to be real actors in humans' lives.

The observation that 'religion' is remarkably absent in the field of post-anthropocentric scholarship in anthropology has been discussed by Nils Bubandt (2018) and Mayanthi Fernando (2022). Bubandt observes that 'studies of the nonhuman remain strikingly and one-dimensionally secular' (ibid.: 7). He considers it of vital importance that 'the breakdown of the human-nonhuman divide destabilizes the distinction between the material and the spiritual, the natural and the supernatural, the skeptical and the superstitious' as well (ibid.: 8). In the same vein, Fernando (2022) argues that fully undoing the nature/culture distinction means attending to the more-than-natural that would still be banished from our epistemological and ontological horizons. She states:

> On the one hand, by extending notions of personhood and agency to other-than- or more-than-humans, scholarship on Indigenous ontologies, new ecologies, and multispecies relations undoes the separation between nature and humanity integral to secularity, an undoing captured by the term natureculture. On the other hand, by delimiting nonhumans to lifeforms conventionally understood as "nature," and by anchoring "nature" in a materialist epistemology and ontology, much of this work also reproduces the separation between natural (coded real) and supernatural (illusory) that was equally integral to secularity. Spirits, gods, and other "supernatural" beings are still excluded from this scholarly conversation as agents, actants, or existents. (ibid.: 568)

Multispecies studies and new animism studies, Fernando argues, tend to develop as an extension of, rather than a departure from, secularity. When scholars grant a more-than-human agency, she states, this 'more-than' is a horizontal, largely egalitarian extension of powers. They thus keep their feet firmly planted on secular ground, as Furani and Robbins (2021, p. 502) also argue. This shows, Fernando states, that the multispecies turn in anthropology remains uncomfortable with the idea of human subjection to another, a hierarchical, inegalitarian, and invisible 'more-than-human' (Fernando 2022, p. 569).

A similar observation was recently issued from the angle of Religious Studies. Kocku von Stuckrad (2023) observes that the academic study of religion hardly responded to contemporary discussions about the nonhuman and interspecies relationships. Scholarship would still reflect patriarchal, colonial, capitalist, and anthropocentric ideologies by not taking other-than-human agency seriously. Even theoretical work that includes nonhuman animals in conceptualizations of religion, he states, still operates in binary constructions of nature/culture, human/nonhuman and body/mind, and thus within a frame that prioritizes the human over the 'rest' of the natural world (ibid.: 2).

In line with the work of von Stuckrad (2023), Fernando (2022) and Bubandt (2018) we believe that including gods, spirits, and other uncanny presences in our perception of more-than-human entanglements is fundamental to our effort to de-secularize and decolonize our anthropological study of more-than-human entanglements. We also consider it a prerequisite to overcome a persistent process of 'othering' that is built on the divide between 'us'—the Western, secular, rational knower—and 'them'—the non-Western, religious, magical believer. Anthropologists are often deeply involved participant observers in the religious lives of the people with whom they work, and consequently write carefully and often sympathetically on the religious lives they studied. Nevertheless, they simultaneously tend to cast doubt on their 'beliefs' to ensure that the borders between a 'secular intellect' and any 'religious tradition' are duly respected (Furani and Robbins 2021, p. 502). To explore how 'the professional thinking of anthropologists as academics could be other, could dwell in non-cognitive homes of wonder, intuition, passions, mysteries, bodies, revelation, or in other spaces generally closed off by secular boundary-drawing' (ibid.: 509), we seek to recreate a professional acknowledgement of wonder and enchantment in anthropology.

To introduce the subject of enchantment and re-enchantment, we briefly return to Max Weber's classical text on "Science as vocation" in which he proclaimed the world to be dis-enchanted:

> The increasing intellectualization and rationalization do not indicate an increased
> and general knowledge of the conditions under which one lives. (. . .) It means
> that principally there are no mysterious incalculable forces that come into play,
> but rather that one can, in principle, master all things by calculation. This means
> that the world is disenchanted. (Weber et al. 1958, p. 139)

'Disenchantment' thus referred to the vanishing of the religious and the sacred from the world as a result of rationalization, secularization and also capitalism, the hallmarks of Western modernity and humanity's progress (Federici 2019, p. 188). Jane Bennett (2010, 2016), however, has argued that practices and experiences of enchantment never disappeared in the modern world. Emphasizing the vitality of nonhuman matter, she challenges the ontological privileging of 'the human' as the only site of vitality and agency. Human agency, she says, is but one within and across a multiplicity of other material bodies and formations. All matter then is very much alive and constituted by a lively and energetic play of forces (2010). In line with Bennett, we challenge the modernity–enchantment contrast by stating that, in the Anthropocene, processes of dis-enchantment may go together with enchanting activities. Following Bennett, we see enchantment as the capacity of human–nonhuman assemblages to mutually generate affective forces and vitalities. In addition to her work, we deliberately extend the focus to the agency and vitality of immaterial actors and explore how they reside in, collaborate with, or defy the material world. This is line with Fernando's (2022) argument to open up for 'higher'-than-human powers but also resonates with Blanes and Santo's (2014) work on 'the social life of spirits', saying that anthropologists have understood gods and spirits not as powerful agents of change but rather as 'symbols' or 'representations' of something else. We also challenge this way of thinking about the spirit world to uncover a livelier and diverse environment of entities interacting with each other.

In this Special Issue we thus deliberately focus on (re-) enchantment as an analytical focus to bring religion back into the discussion on the nature/culture divide, and to re-value in our anthropological study alternative ways of both enchanting nature and of being enchanted by nature. We follow scholars from various other disciplinary backgrounds who also make a strong case for a re-rehabilitation of enchantment in the discussion on more-than-human socialities in the Anthropocene (e.g., Bennett 2010; Curry 2012; Gibson 2009; Rangarajan 2022; Plumwood 2002a). These scholars also contest the idea that the experience of enchantment belongs to past ages of 'superstition' and accordingly has to be condemned as 'irrational', 'primitive' or 'childish' (Haberman 2020). James Gibson (2009, p. 12), for example, defines re-enchantment as 'a new quest for kinship with nature' that 'refuses to reduce nature to a set of objects infused with utilitarian purpose'. Swarnalatha Rangarajan (2022, p. 64) stresses the 'relevance of reconceiving nature as innately innovative, spiritual and poetic' and states that 'by creating new alter-tales of human kinship with more-than-human others, we challenge the narratives of disenchantment'. Also Patrick Curry (2012, p. 78) states that 'while being in a non-instrumental relationship with nature, it matters to recognize nature's innate capacity to enchant' and 'to find presence, sentience, and responsiveness to our human agency in the material other-than human world'. Moreover, Plumwood agrees that enchantment is rooted in the more-than-human nature 'as it is living and sensuous', and its materiality is 'already full of form, spirit, story, agency, and glory' (Plumwood 2002b, p. 226).

As co-editors, we consider the possible (re-)enchantments of nature as an entry point to think beyond the nature–supernature divide, and to simultaneously include a gender perspective on non-divided 'uncanny ecologies' (Fernando 2022). Our focus on gender matters here as particularly women's embodied religiosity—which often includes their sensory and poetic practices of enchanting the more-than-human—tends to be overlooked, or even denied the status of 'religion', when viewed from a Western Christian, colonial and male rationalist perspective (e.g., Notermans 2019; Notermans and Swelsen in this issue). When in more-than-human entanglements different actors mutually touch, energize and speak to each other, a gendered nature/supernature may speak or act differently to human gendered bodies. We wonder if the crossing of the nature/supernature binary

in humans' lived religious experience and the confluence of energies that emerge from (re-) enchantment capacities work out differently for men and women, and how ecological responsibilities and practices of maintenance, and nurturance are differently gendered, in both 'nonwestern' and 'western' (secularized) contexts. Together, these contributions represent a wide-ranging collection of thoughtful yet questioning ethnographic studies on the intersection of gender, nature and religion.

## 4. The Contributions

Taking an integral perspective on the volume, we draw three overarching, shared insights from the nine contributions to this Special Issue. First, human–nature relatedness often springs from the intimate, strongly material interactions with the environment, which is, just as humans, understood to be experienced as divinely gendered. Due to the influence of societal and environmental changes these relations are, however, increasingly under strain, changing not only livelihoods but also the affective connections across the species.

Related to this, our second insight drawn from the collection is that the pressure of current anthropogenic changes in the environment is differently borne by men and women. In understanding how gender is connected to and organizing a diverse more-than-human world, we note that the power, or the lack thereof, humans and nonhumans find in their relatedness are closely tied to questions of socio-ecological justice. The intricately coupled forms of social and ecological justice are particularly effecting the position of women, frequently enhancing their (societal) vulnerability.

Thirdly, as a counter response to the current environmental crises, we see that specific forms of care as material and religious means of communication is what enchants human–nonhuman relationships. In other words, to learn about forms of re-enchantment in the Anthropocene, we suggest focusing on how practices of care may (re-)kindle and vitalize more-than-human worlds, and reversely, inspire human living to be touched by more-than-human and supernatural beings with whom we share our environment. In what follows, we will show how the contributions speak ethnographically to these three insights in different ways.

### 4.1. Gendered Nature and Relationality

The relations that humans maintain with nature and how these come to be expressed through divinely gendered properties and capacities of both humans and land are especially well shown in the papers by Catrien Notermans and Luna Swelsen, Tara Bates, and Annemiek Prins. Their studies, all situated in South Asia, describe how gender defines human–nature understandings and equally shapes their (collaborative) interactions with nature. Put otherwise, women nurture different relations with nature than men, because the more-than-human and supernatural in nature itself are gendered. A comparative, diachronic review by Veronica Strang clearly supports this observation, as she shows how through time, conceptions of gender continued beyond human relations into social connections and partnerships with the nonhuman world.

In the contribution of Notermans and Swelsen, we learn about Hindu women's lived relatedness to the land in Udaipur, India. The described women affectively nurture and enchant the land and soil in ritual action and daily care as a sentient, agentive, divine female companion, aiming for mutual prosperity for human and nonhuman kin. Precisely this religious and gendered intimacy, the authors argue, is what is generally overlooked in the Indian state policies directed at promoting gender equality in land rights and economic development. In the feminist striving for land rights, land is ultimately seen as a resource for income and empowerment, thus ignoring socially respected and valued gender complementarity embedded in multispecies relations. Indeed, these liberal feminist views contrast sharply with the religious priority the women themselves give to bonding with land as both a nourishing nonhuman person and powerful goddess.

A similar relevance of the role of gender in the organization of human–nature relations can be seen in the article by Bates. In her study on multispecies co-inhabitation in pastoral

areas of Tibet, she critically analyzes the common discourse on domestication, showing that it should be understood as a mutual practice in which women, bovines, the environment and spiritual entities give shape to the domus, and where none has precedence over the others. This is not to ignore the influence of political and economic changes that have an impact on traditional pastoralist practices of the Tibetan Limey herders. In particular, the combined influence of the historical annexation of pasturelands by the People's Republic of China as well as the feminization of herding because of seasonal migration of the men as wage workers have blurred gendered identities and practices. That is, only single and widowed women take the cattle into the mountains, married women stay with their men in the village and focus on agriculture. Through the women's shared gender with domestic animals, the traditional, patriarchal norms that structured both human and multispecies relations are affirmed in the villages, accordingly socially marginalizing alternative human–nature relations that the herding women maintain.

A likewise challenged and gendered burden in taking care of the household, and dealing with more-than-human agents, is described in an article by Prins. She focuses on the spiritual and gendered experiences of dwelling-in-uncertainty in the context of Bengali char lands. As with the feminine herders in the study of Bates, owing to seasonal labor migration of the men, it is the women who have to both move with and resist the water that as a result of climate change, becomes increasingly unpredictable and risky. Even so, Prins points out that char communities have always lived with the energetic and erratic agency of the river, which is believed to harbor divine, immaterial and spiritual powers. Over time, women have played a crucial role in fostering a sense of continuity in the fluidity of the amphibious landscape, shifting and resettling homesteads on the sand banks. It is, however, this already demanding balancing act of engaging with a landscape, which is only knowable to some extent, that has come under strain due to climate change, increasing the velocity of the cycles of loss and return and, thus, the care work of women.

*4.2. Intersections of Gender and Socio-Ecological Justice*

The second overarching insight of this Special Issue, specifically the interconnectedness of gender and issues of socio-ecological justice, is particularly demonstrated in studies by Karolien van Teijlingen, Veronica Strang, and Sina Pfister and Edwin de Jong. Strang's paper speaks, moreover, to historically ingrained and contemporary societal structures of dominance and inequality that affect both gender and nature and are often authorized by dominant religious narratives. Her analysis grounds the work by van Teijlingen and Pfister and de Jong, and supports the point that Ortner and d'Eaubonne already made in 1974, which is still echoed by current-day ecofeminists. That is, the oppression and social marginalization of women are intimately connected to the likewise exploitation of the earth. Consequently, the one can therefore not be resolved without acting on the other as well.

Van Teijlingen addresses in her paper the role of the Catholic church in large-scale mining in Ecuador. She notes how from the 1960s onwards, efforts were made by progressive liberation and eco-theologians to reposition the Catholic church as 'the Church of the Poor' to connect Catholicism to the struggles for environmental justice in the conflict-ridden mining landscape. However, despite the celebrated emphasis on both indigenous people and women in the Church's new green agenda, the actual emancipatory influence may be limited, precisely because of the layered connections between their territory and indigenous, female bodies. Feminist demands for autonomy of reproductive rights, radical action against gender-based violence, and female leadership do not sit well with traditional gender roles in the Church as well as in the broader patriarchal society.

The case study described by van Teijlingen confirms Veronica Strang's point on the inability of most world religions to foster the emancipation of both nature and women. In Strang's comparative overview she shows that there are diverse historical trajectories in how human relationships with the nonhuman world evolved. She addresses especially how larger societies have undergone major transitions, in which changes in religious beliefs and practices cojoined social, political and economic transitions. In traditional placed-based

societies, such as in Indigenous Australia, conviviality, care and interdependence across species have always been key values, which went hand in hand with the recognition of gender equality in all domains. In the era of imperialism and colonization, Strang observes, however, a dual movement of growing patriarchism, and an ensuing subjection of women, together with a religiously disenchanted natural world, over time emptied of its pluralistic deities, ancestors and superhuman beings. Like the social position of women, the material world is thus recast as a passive subject that can readily be exploited for human purposes.

In the patriarchal monotheistic ideology described by Strang, the parallel between nature and women is generally associated in a negative way, justifying and reproducing inequality and a lack of rights. The case study of Pfister and de Jong draws attention to an alternative discourse, away from the history of patriarchal colonization and submission of both women and nature into capitalist, extractivist economies. Resonating with the paper by van Teijlingen, they focus on how compounding environmental crises in Constitucíon, Chile, have set in motion a change in thinking about human–nature entanglements. A reappraisal of the spirited concept of Mother Nature, grounded in indigenous Andean cosmologies, aligns with an often female activism for conservation and restoring (local) biodiversity. Moreover, as in Ecuador, the struggle for environmental justice is coupled with action for more social justice and recognition of women's rights and their public position. Yet, rather than striving for emancipation by blending in with the male stereotype, women recognize and claim their power through their traditional roles, and emphasize feminine notions of care, attentiveness and sensitivity as transformative capacities for enhancing environmental protection.

*4.3. Reciprocal Pathways between Care and Enchantment*

In line with the above insight, we suggest as our third finding that care and care practices form a key reciprocal pathway between humans and the nonhuman world that emerges through rituals and experiences of enchantment. Meaningful, sensitive and gendered human–nonhuman relationships come to be organized through care. The positive, affirmative connection especially between women and care as transformative both for the environment and socio-political relations is touched upon by other authors (see Notermans and Swelsen, Westendorp and Gould, and Madrell in this issue). The papers in this issue point out that the nonhuman, sentient, spirited world is equally agentive in enacting meaningful relations across the species. For instance, the Hindu women in the article by Notermans and Swelsen (this issue) know that literally enchanting the land is necessary in order to vitalize the land and soil, and to be able to enter into meaningful nourishing relations. Put differently, humans, but especially women, enchant and are being enchanted, and so come to be included in a multispecies web of relations. Recognition of this entanglement further incites action to battle against current environmental crises. The contributions by Mariske Westendorp and Hannah Gould, and Avril Madrell, Irene Becci and Alexandre Grandjean show the enabling force of enchantment as well as the desire to be enchanted.

Westendorp and Gould focus on how gender plays a role in remaking Western death care. Discussing two examples of the Natural Death Movement (NDM), the work of 'death doulas' and the emergence of arboreal necro-technologies, respectively, they highlight the usage of fertility symbolism and the renewal of life through death; capacities that are closely associated with feminine reproductive and care practices. Within NDM, they notice, moreover, a religious and spiritual re-invention that taps into a re-enchanted nature. They argue that a feminine approach in care for the dying and the dead is perceived to be a possible solution to the ecologically damaging practices of what they term the masculine Death Industrial Complex.

A comparable re-enchantment is noticeable in the case studies discussed by Becci and Grandjean and Madrell. The first show an activist process of spiritualizing ecology. Less based in traditional religious faith, but rather drawing on New Age spirituality and Neo-Shamanism, the authors describe different forms of eco-activism in Switzerland that

are increasingly attended by women and steered by female spiritual leaders, such as the so-called holistic milieu and the transition network. Some eco-activists put forward a critical focus on gender, arguing for the recognition of gender diversity beyond singular categories of male and female, stating the need to mirror the gender complementarity that also exists in nature. Other spiritual groups call for a 're-enchantment of the whole world', connecting environmentalism to a sacralized nature.

Madrell, finally, describes Celtic pilgrimages to the British Isle of Man, where the walks are framed by the narratives and ethos of both medieval and contemporary nature-centered expressions of Celtic spirituality. She suggests how pilgrims, predominantly women, through an interplay of embodied movement, nature and spiritual experience find inspiration, and, in fact, become re-enchanted, to develop collective faith-based environmental action. In the face of the uneven negative impacts of the Anthropocene, pilgrimage can thus offer an educative way of exploring environmental repentance and expressing faith-based solidarity with the world beyond the human, and commit to alternative, caring forms of connectedness.

### 4.4. Conclusions

In this Special Issue, we have brought together a multidisciplinary collection of ethnographic studies in an editorial effort to bring back into analytical focus the embodied practice of gender and religion in contemporary discussions of the often fraught human–nature relations in the Anthropocene. Taking up the critical and foundational work of early feminist scholars and anthropologists, we asked in what ways gender is formative in the mutual relationship between humans and nonhumans. The contributions show that across the world and through time, nature is seen to be gendered, and accordingly gives shape to the multispecies relations and interactions along gendered lines. Moreover, what is considered appropriate for different genders in their contact with the environment is often religiously informed, guiding, in particular, women's work with nature towards a ritualized behavior of sensitive, careful respect. The conceptual and religiously sanctioned intertwining of women and nature influences contemporary strivings for socio-ecological justice. The parallel between patriarchal subjection and marginalization of women and the exploitation of the earth, fiercely criticized by ecofeminist scholars and earlier also by anthropologists, most notably Ortner, continues into the present. The fate of the earth is inextricably associated with that of the emancipation of women, indigenous peoples, and other minorities. Even so, the close connection between lived caring relations and the enchanting capacities of nature points to an acknowledgement of what we propose is a sentient, agentive environment that actively invites and extends a plurality of entanglements between humans and nonhumans.

We suggest that the proceeds of this Special Issue show the relevance of enlarging our analytical interest to the tacitly religious, spiritual and often marginalized yet vital gendered human–nature relations. This may not be an easy task, as it also asks us to recognize the material vibrancy and agency of the more-than-human world, something that may feel uncomfortable in a tradition of science that was born from a clear nature–culture dichotomous way of thinking. This difficulty and the neglect, even if unintentional, of the influence of gender and religion in anthropogenic environmental crises were reflected in the realization of this Special Issue. Although the call for papers was broadly welcomed as relevant and scientifically urgent, the threefold integration of gender, nature and religion turned out to be a challenge for the authors. The majority of the contributors had not looked at their research data from this combined perspective. As editors, we thus found ourselves encouraging them to become aware of the analytical blind spots they might have and to revisit their material with attention to both gender and religion. We feel that their efforts and stamina were worth it. The case studies reveal an ethnographic richness on the central topics of this issue, illustrating persuasively the value of integrating gender and religion into future anthropological scholarship on the Anthropocene.

**Author Contributions:** Ideation, C.N. and A.T.; literature search and analysis, C.N. and A.T.; writing and revision, C.N. and A.T. All authors have read and agreed to the published version of the manuscript.

**Funding:** This research received no external funding.

**Data Availability Statement:** No new data were created or analyzed in this study. Data sharing is not applicable to this article.

**Conflicts of Interest:** The authors declare no conflict of interest.

## Notes

1    From different angles the term Anthropocene has been ardently debated in social sciences which resulted in alternative concepts like Capitalocene (Moore 2016), Chthulucene (Haraway 2016), or Carbocene (Lecain 2015). Though we empathize with these critical conceptual nuances, we adhere to the original term Anthropocene for reasons of consistency in this Special Issue.

2    See Baehr (2017) for a differentiating account of western liberal feminism.

3    See Roa (2012) for a differentiating account of ecofeminism.

4    An exception to this observation is the work of the Australian ethnographer Deborah Bird Rose. She closely worked together with Val Plumwood (see Rose 2013) and both contributed to the interdisciplinary volume *Feminist Ecologies* (Stevens et al. 2018), edited by Lara Stevens et al.

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
