# Peer review of "Revaluing Gender and Religion in the Anthropological Debate of the Anthropocene: A Critique on the Threefold Culture–Nature–Supernature Divide"

_religions, doi:10.3390/rel15020218_

Round 1

Reviewer 1 Report

Comments and Suggestions for Authors

This editorial to the issue "Revaluing gender and religion in the anthropological debate of the Anthropocene: A critique on the threefold culture-nature-supernature divide" is very well written. The theme and objective  are clear; it addresses a particular gap in theory and research in the study of religion (specifically in the discipline of anthropology ) and offers a state of the art. It introduces a series of nine articles related to the theme but cover a great variety of research in topics and geography. The articles are introduced according to a three-fold structure. It is also clear that the editors have worked with the contributors (or some of them) to rethink and rewrite their work in view of the shared theme. (gender-religion; nature-culture). One remark is that I do miss some more reference to work in the anthropology or ethnography of gender in religious and/or spiritual communities that does address topics of nature/culture; disenchantment; posthumanism (or applying more new materialist methodologies etc). There is certainly more work out there; although this does not directly engage with the question of a there being a 'gap' in the anthropological debate in the study of the Anthropocene. Perhaps the editors focus very clearly on those who situate themselves more strictly within the discipline of anthropology or publish in anthropological journals? I would argue there is much more fluidity and interdisciplinarity within the field of gender and religion generally; many in that field use ethnographic methods and take inspiration from posthumanist debates that go beyond and problematize and secular-religious and nature-culture binaries (among them also sociologists, theologians, gender studies scholars, etc), whether or not they might engage with mainstream anthropological debates.   Some general references as suggestions: the two volumes by anthropologists A. Fedele and K.E. Knibbe (Gender and power in contemporary spirituality and Secular societies, spiritual selves: the gendered triangle of religion, secularity and spirituality) and the general reader The Routldege Handbook of Religion, Gender and Society. But otherwise, I enjoyed this editorial and look forward to the full issue! 

Author Response

We thank you for your constructive feedback on our work. We responded to your comment by adding the following sentence at the start of our article:

"Of course we do acknowledge that interdisciplinary work exists with an ethnographic focus on gender and nature in religious contexts (e.g. Fedele and Knibbe 2013, 2020, Starkey and Tomalin 2022), yet these do not directly engage with the socio-ecological impact and the changing gendered relations the Anthropocene generates."

We also added the three recommended publications in our references.

Reviewer 2 Report

Comments and Suggestions for Authors

My comments on this essay will be brief. This is the introductory essay to a special issue that seems to be timely and well-conceived. The authors are in command of their field (anthropology), the history of the disciplines and issues they discuss (especially feminist theory), and appear to have assembled a strong group of contributors. I found the essay instantly and easily persuasive as to the significance of the subject they address (the revaluing of gender and religion in the critical discourse of the Anthropocene), clear in its argument and implications, and helpful in the history it reconstructs (e.g., of second wave feminism and the nature-culture divide it inherited and reinforced).

The essay reads like the thoughtful work of scholars who are highly informed in the scholarly fields they are addressing. The timeliness of their work also strikes me as exemplary, less as regards the Anthropocene per se than as regards the oversight (to this point) of the dynamics of gender and religion in most arguments that evoke the Anthropocene.

I have no real suggestions for improving the manuscript. It's clear, thoroughly researched, and well written. I'm delighted that Religions is the forum for this special issue and think it will further cement the journal as a frontline venue for important conversations that reveal the significance of religion to a host of issues that inform our vital but fragile world.

Author Response

We thank you for your positive feedback which makes us feel even more encouraged to publish our work.